# Guillain-Barré Syndrome with Incomplete Oculomotor Nerve Palsy after Traumatic Brain Injury: Case Report and Literature Review

**DOI:** 10.3390/brainsci13040527

**Published:** 2023-03-23

**Authors:** Jinsheng Liu, Feng Tang, Xinjun Chen, Zhiqiang Li

**Affiliations:** Department of Neurosurgery, Zhongnan Hospital, Wuhan University, Wuhan 430062, China

**Keywords:** Guillain-Barré syndrome, traumatic brain injury, electromyography, case report

## Abstract

Guillain-Barré syndrome (GBS) is a severe peripheral neuroinflammatory demyelinating disease characterized by symmetrical progressive limb weakness, which can be accompanied by cranial nerve and sensory disturbances. There is usually a history of bacterial or viral infection prior to onset. GBS is rarely seen after traumatic brain injury (TBI). We report a case of a 66-year-old male patient who presented with dilated pupils, followed by respiratory failure and symmetrical quadriplegia during a conservative treatment for TBI. He was eventually diagnosed with GBS and was treated with intravenous immunoglobulin, followed by rehabilitation therapy with a good recovery. We summarize previous similar cases and analyze possible causes. It is suggested that the possibility of GBS should be considered when unexplained symptoms occurred in patients with TBI, such as respiratory failure, dilated pupils, and limb weakness.

## 1. Introduction

Guillain-Barré syndrome (GBS) is an autoimmune-mediated peripheral neuropathy that mainly affects most spinal nerve roots and peripheral nerves, often involving cranial nerves. It is characterized by symmetrical limb weakness with sensory and cranial nerve damage. The syndrome is usually preceded by a history of prodromal infection, such as Campylobacter jejuni, cytomegalovirus, Epstein–Barr virus, etc. [1] The correlation between GBS and post traumatic brain injury (TBI) was first reported in 1987 [2]. A subsequent study has reported that surgery and trauma were possible triggers of GBS and trauma patients with surgical treatment exhibited a higher risk of developing GBS than those without [3]. Some recent studies also presented similar results, indicating TBI may be a risk factor of GBS [4,5,6]. Here, we report a case of GBS with incomplete oculomotor nerve palsy after traumatic brain injury.

## 2. Case Presentation

A 66-year-old man was admitted to the emergency department because of a fall injury. On admission, he was drowsy, with vomiting and urinary incontinence. His Glasgow Coma Scale score is 12 (eye opening: 3 scores; verbal response: 4 scores; motor response: 5 scores) and he had no history of any infections, hypertension, diabetes, or heart disease.

The craniocerebral computed tomography (CT) examination on admission showed a contusion of the bilateral frontal lobe, multiple hematomas in the bilateral frontal lobe and right temporal lobe, and fractures in the right frontal bone and bilateral parietal bone. Subarachnoid hemorrhage (SAH) was considered because the density of the falx cerebri, bilateral tentorium cerebelli, and some sulci were increased (Figure 1). A cervical vertebra CT scan showed no abnormalities.

The patient was transferred to the neurological intensive care unit (NICU) for observation. Supportive treatments were performed such as dehydration, analgesia, and antiepileptic drugs, and regular craniocerebral CT scans. On the third day of admission, the patient was steady with a Grade 4 muscle strength in his extremities and discharged from NICU. On the fifth day of admission, the patient’s condition became worse. His consciousness state changed from alert to drowsy again. The muscle strength of the left and right limbs was Grade 3 and Grade 2, respectively. An emergency head CT (Figure 2A,B) showed no new intracranial rebleeding, and a low-density edema zone was observed around the hematoma. On the ninth day, the patient’s right pupil was suddenly dilated with a 5 mm diameter, with a disappeared light reflex, while the left pupil was normal. An urgent craniocerebral CT scan (Figure 2C,D) only showed that the extent of the edema was enlarged with no intracranial rebleeding. The dehydration treatment was intensified.

Ten days following admission, however, bilateral dilated pupils with a disappeared light reflex were observed. However, the patient could open his eyes under verbal stimulation and localized pain. Although the CT scan (Figure 2E,F) revealed little changes, a bilateral frontal hematoma evacuation plus decompressive craniectomy were performed. The postoperative CT was shown in Figure 3. The patient’s symptoms continued to worsen after surgery. There were no changes in the bilateral dilated pupils. In addition, he developed respiratory failure and mechanical ventilation was required on the first postoperative day. In addition, the motor disturbance of the extremities occurred on the second postoperative day. The muscle strength of the extremities became Grade 0. A pulmonary CT scan indicated the pulmonary infection and massive pleural effusion. Then, a closed drainage of the bilateral chest was performed. However, he remained conscious and experienced the pain sensations. The possibility of GBS was suggested and the lumbar puncture was subsequently performed.

The analysis of the cerebrospinal fluid (CSF) showed a total protein of 1.0 g/L and nuclear cell count of 12/µL. This indicated the albuminocytological dissociation. Anti-ganglioside antibodies (only anti-GD3 antibodies) were detected to be positive in the patient’s serum but not in his CSF.

Intravenous immunoglobulin (IVIg) at 0.4 g/kg per day was given to this patient for 5 days. The patient’s condition improved slightly after a course of immunoglobulin therapy, and the electromyography (EMG) at this time showed a prolonged latency, decreased amplitude, and absence of F waves in multiple pairs of nerves in the extremities, which indicated the peripheral nerve damage (including demyelination and axonal damages) in the extremities, involving the motor fibers, the sensory fibers, and the proximal nerve roots. Table 1 shows the motor and sensory nerve conduction studies of the patient’s extremities. A craniocerebral CT scan during this period was shown in Figure 4. After the IVIg treatment and rehabilitation treatment, the patient slowly recovered.

One month later, the patient was discharged from the ventilator, and the muscle strength of his extremities recovered to Grade 4. This further confirmed the diagnosis of GBS.

## 3. Discussion

We report a case of a patient with GBS after TBI who developed bilateral mydriasis on the tenth day of conservative management. Clinicians lack awareness of GBS after TBI. R. Duncan et al. first reported a case of GBS after craniocerebral trauma in 1987 [2]. Since then, there have been a few reports of GBS after traumatic brain injury in the past three decades. We have made a summary of the clinical characteristics of all these cases as detailed in Table 2. Most of these patients were middle-aged and elderly, with an average age of 52.5 years old (ranging between 22–75). There were nine men and four women, and all were admitted because of TBI. Our reported case was a 66-year-old man with moderate craniocerebral injury. His head CT scan showed worsening edema in the bilateral frontal and right temporal lobes.

The interval between the leading event and the onset of GBS was mostly one to two weeks (69%). In the previous 13 case reports, their initial symptom is usually motor and/or sensory disturbance of the extremities during treatment. Two patients had dilated pupils [8,10], but only one patient presented the dilated pupils as the first symptom [10]. The patient presented with dilated pupils on the seventh day of admission, at which time he was conscious in bed and could be positioned. However, his conditions deteriorated dramatically within 30 min, with the absence of brainstem function, quadriplegia, the absence of deep tendon reflexes, and paralysis of respiratory muscles. He was eventually diagnosed with fulminant GBS. On the fifth day after injury in our case, the bilateral muscle strength of the limbs was decreased. On the ninth day, the diameter of a single pupil was suddenly dilated. Subsequently, bilateral dilated pupils, the decreased muscle strength of the extremities, and respiratory failure were observed.

The role of CSF testing and electrophysiological examination in the diagnosis of GBS is very important. Except for the case reported by Duncan and colleagues in 1987 without a lumbar puncture examination [2], all cases showed albuminocytological dissociation in CSF. An electrophysiological examination showed a prolonged latency of motor and/or sensory nerve conduction, decreased amplitude, and absence of F waves. The examination results of our case were consistent with the reports. Moreover, anti-ganglioside antibodies are closely associated with GBS. A previous study showed that the presence of anti-GD3 antibodies was associated with oculomotor dysfunction [18]. In the serum of our reported case, the anti-GD3 antibody was also detected. This may explain why the patient developed pupillary symptoms.

GBS is usually preceded by a pro-infection event, and molecular mimicry between nerves and microbial antigens has been demonstrated to cause GBS [19]. However, the etiology of GBS after TBI is still inconclusive. Generally, TBI included two phases, and the first phase of TBI is caused by a direct mechanical injury, while the second phase is induced by systemic pathophysiological changes. In the first phase, the stress reaction caused by trauma activates some underlying or subclinical process, leading to changes in the body’s immune response, which might be one reason for GBS [20,21]. In the second phase, previous studies have shown that TBI could mediate the activation of B cells and the production of auto-antibodies, which might be pathological and cause GBS [22]. Ankeny et al. [23] found that mice lacking B cells achieved better motor recovery and reduced pathological manifestations compared with wild-type mice in a TBI model, which also supported the notion that TBI-induced B-cell activation is pathological. In our current case, the GBS more likely occurred during the second phase of TBI. After TBI, the blood–brain barrier is destroyed, and specific proteins and decomposition products in the brain enter the peripheral circulation through CSF as auto-antigens. These auto-antigens are exposed to the immune cells, which, therefore, activates B lymphocytes to produce auto-antibodies. Among these auto-antibodies, ganglioside antibodies are closely related to the occurrence of GBS [1]. Studies have shown that 50–60% of patients with TBI or spinal cord injury will produce a series of central nervous system protein antibodies including ganglioside antibody GM1 [24,25]. Anti-GD3 antibodies were detected in the serum of our reported cases. However, we notice that not all patients with TBI will develop GBS, which may be associated with the type and amount of auto-antibodies released after TBI. We speculate that taking the ganglioside antibody in serum as a routine examination for TBI patients may be meaningful for the prevention, monitoring, and early treatment of GBS after TBI.

There are still challenges in dealing with the TBI-accompanied GBS because of its rarity. Clinically, the pupil changes and the motor dysfunction are the important indicators when considering the necessity of surgery. Intracranial rebleeding or the aggravation of the cerebral edema can easily lead to the deterioration of the condition, resulting in changes in the patient’s consciousness, pupils, and muscle strength. At this time, it is difficult for us to associate the patient’s bilateral dilated pupils with GBS, because it is a manifestation of a rare symptom of GBS. This may affect the doctor’s judgment, delay the treatment of the disease, and lead to a poor prognosis. For our reported case, we were entangled in whether the patient needed to undergo a hematoma evacuation and decompressive craniectomy. The patient also had respiratory failure on the first postoperative day, which may have been related to the patient’s pleural effusion and hypostatic pneumonia in the lungs. In fact, it now seems possible that the respiratory failure also resulted from the paralysis of the respiratory muscles caused by GBS. However, it was not until the patient developed symmetrical quadriplegia on the second postoperative day that we considered the diagnosis of GBS and performed an examination of the CSF. This is the experience and lesson that we should learn in clinical diagnosis and treatment.

## 4. Conclusions

Clinicians should consider the diagnosis of GBS when quadriplegia, respiratory failure, or pupil changes occur without an obvious reason after TBI or surgery. At this time, a lumbar puncture and electromyography are necessary, which help us to confirm the diagnosis and carry out the effective treatment. The routine examination of serum ganglioside antibodies may be beneficial for the monitoring and early treatment of GBS after TBI.

## Figures and Tables

**Figure 1 brainsci-13-00527-f001:**
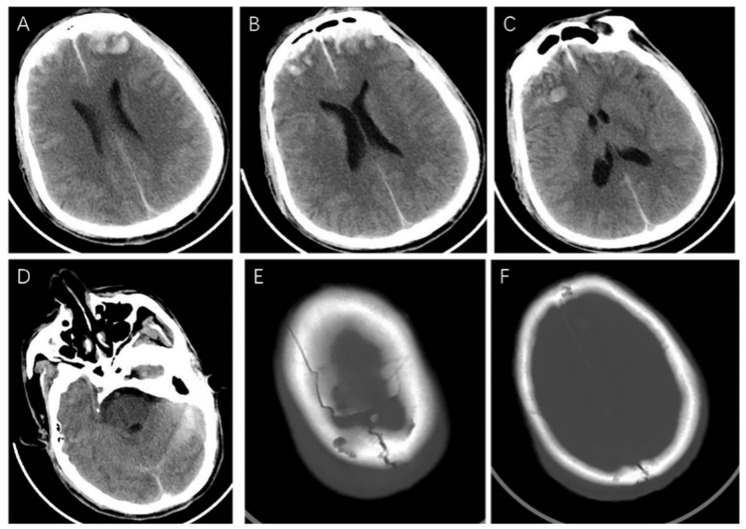
The craniocerebral CT examination on admission. It showed (**A**–**C**) contusion of bilateral frontal lobe, multiple hematomas in bilateral frontal lobe and right temporal lobe, (**D**) subarachnoid hemorrhage, and (**E**,**F**) fractures in right frontal bone and bilateral parietal bone.

**Figure 2 brainsci-13-00527-f002:**
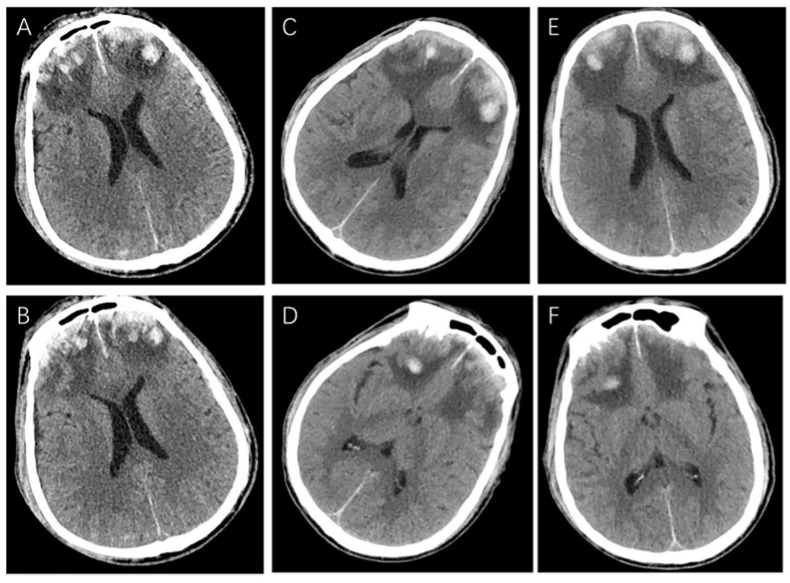
The craniocerebral CT examination during conservative treatment. The craniocerebral CT examination on the fifth day of admission showed a low-density edema zone around the hematoma (**A**,**B**). The craniocerebral CT examination on the ninth day of admission showed the extent of edema was enlarged (**C**,**D**). The repeated CT examination on the tenth day of admission showed little changes (**E**,**F**).

**Figure 3 brainsci-13-00527-f003:**
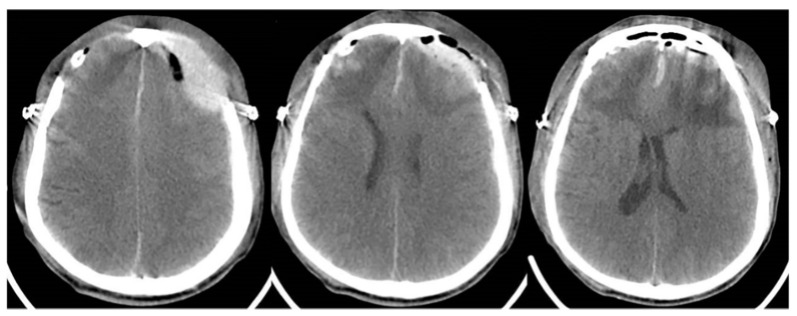
The craniocerebral CT examination on the first postoperative day. The bilateral frontal bones showed postoperative changes, with gas accumulation in the operative area, reduced hematoma range, and little change in the surrounding low-density edema zone.

**Figure 4 brainsci-13-00527-f004:**
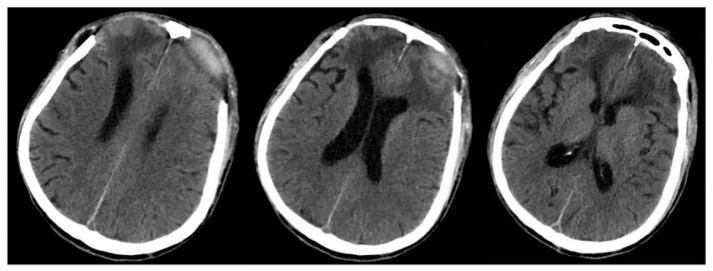
The craniocerebral CT examination on the eighteenth postoperative day, after bilateral frontal decompressive craniectomy. The bilateral frontal lobe brain tissue was slightly swollen, and the edema was better than before.

**Table 1 brainsci-13-00527-t001:** The result of electromyography examination.

Nerve	Site	Latency (ms)	Amplitude (mV)	Conduction Velocity (m/s)	Distance (mm)
MNCS					
Left ulnar	Wrist	3.87	1.62		45
	Below elbow	9.35	1.47	42.9	235
	Above elbow	11.20	1.41	45.9	85
Right ulnar	Wrist	5.54	0.75		30
Left median	Wrist	13.50	0.40		45
	Elbow	20.20	0.33	33.6	225
Right median	Wrist	14.20	0.50		50
Left tibial	Ankle	8.82	0.63		80
Right tibial	Ankle	8.00	0.92		90
Left peroneal	Ankle	No response			
	Bl Fib.head	No response			
Right peroneal	Ankle	No response			
	Bl Fib.head	No response			
SNCS					
Left ulnar	Wrist	No response			
Left median	Wrist	No response			
Right median	Wrist	No response			
Left superficial peroneal	lower leg	1.82	9.80	44	80
Right superficial peroneal	lower leg	2.02	2.90	37.1	75
Left sural	middle leg	1.33	7.50	56.4	75
Right sural	middle leg	No response			

MNCS: motor nerve conduction studies; SNCS: sensory nerve conduction studies; Bl Fib.head: below fibula head.

**Table 2 brainsci-13-00527-t002:** Summary of the reported cases with Guillain-Barré syndrome after traumatic brain injury.

Author/Year	Age	Gender	Activating Events	Interval between Activating Events and GBS	Clinical Features	CSF Examination	Electrophysiology
R. Duncan et al., 1987 [2]	61	male	TBI after falling off a ladder, brain contusion, and subdural hematoma.	15 days	Tetraplegia, dysphagia and dyspnea, and bilateral lower motor neurone facial palsies.	Not performed.	Generalized predominantly motor demyelinating peripheral neuropathy.
De Freitas G R et al., 1997 [7]	29	male	TBI after hitting the head, and subarachnoid hemorrhage.	7–9 days	Tetraplegia, facial diplegia, deep areflexia, and respiratory failure.	Increased protein concentration.	Increased latency, part of the amplitude was reduced, and no F waves.
Stojkovic, T et al., 2001 [8]	47	male	TBI, skull fracture, extradural hemorrhage, and multiple intracerebral hematomas. He underwent evacuation of epidural hematoma.	8 days	Tetraparesis, mydriasis, and respiratory failure.	Protein: 1.97 g/dL; cell count: 2/mm^3^.	The latency was prolonged, the amplitude was shortened, and the F-wave was absent.
Lin, Tsai-Ming et al., 2006 [9]	22	female	TBI after a motor vehicle accident, and facial bone fracture.	10 days	Weakness in all 4 limbs, most severe in the lower extremities, and numbness of both lower legs.	Acellular with a protein of 0.5 g/L.	Absent of F-waves, revealed a severe generalized predominantly motor demyelinating peripheral neuropathy.
Rivas, Sharon et al., 2008 [10]	55	male	TBI after falling during an alcohol-withdrawal-related seizure, parietal bone fractures, brain contusion, and subdural hematoma	7 days	Absence of brainstem function, flaccid quadriplegia, absent deep tendon reflexes, and respiratory failure.	Increased protein level with normal cell counts.	All nerves and fibrillation potentials of all muscles are inexcitability.
Yardimci, Nilgul et al., 2009 [11]	75	female	TBI after a traffic accident, and subdural hematoma	7 days	Quadriparesis, choreic movements in the right arm, and cerebellar ataxic speech.	Acellular with a high protein level.	Severe generalized, predominantly motor-demyelinating peripheral neuropathy.
Tan, Ik Lin et al., 2010 [12]	44	male	TBI after a micturition syncope episode, skull fracture, brain contusion, and subarachnoid hemorrhage	7 days	Areflexic tetraplegia, dysarthria, dysphagia, and respiratory failure.	Albuminocytological dissociation, and the protein level is 1.82 g/L.	Absent motor and sensory responses, absent blink reflexes, and an absence of spontaneous and voluntary activity on electromyography.
Battaglia, F et al., 2013 [13]	73	female	TBI after falling from a stool, cerebral hemorrhage and lumbar vertebrae fracture, and spinal surgery	7 days	Dysphagia, asymmetric facial diplegia, the muscle strength of the limbs decreased, and deep and superficial hypoesthesia of both legs.	Albuminocytological dissociation.	An increase of motor distal latencies and temporal dispersion of motor action potential in four limbs, and F-waves latencies were increased in lower limbs.
Zhang, Guan-Zhong et al., 2014 [14]	56	male	TBI after falling from height, epidural hematoma, and brain contusion	Not mentioned	Tetraplegia, and hoarseness.	Protein level was increased without alteration in cell numbers.	Motor nerve conduction velocity was reduced and F-wave latency was extended.
Carr, Kevin R et al., 2015 [4]	58	male	TBI after being struck by a fallen branch, brain contusion, subarachnoid hemorrhage, subdural hematoma, and facial bone fractures	17 days	Bilateral lower extremities and face weakness	Albumincytologic dissociation.	Conduction block in both upper and lower extremity motor nerves, and the latency of F wave was prolonged.
Li, Xiaowen et al., 2017 [15]	48	female	TBI, unspecified.	10 days	Weakness on both limbs, deep tendon reflexes were absent, and respiratory muscle involvement.	Albuminocytological dissociation, and the protein level is 0.64 g/L.	CMAP amplitude reduction, and no F waves.
Yonekura, Satoru et al., 2018 [16]	74	male	TBI after falling on a mountain, subdural hematoma, and subarachnoid hemorrhage	3 days	Quadriplegia, bulbar palsy, and weakness of respiratory muscles.	Protein: 92 g/dL; cell count: 8/mmc.	Not performed.
Yilmaz, Hakan et al., 2020 [17]	41	male	TBI after gunshot wound, subdural hematoma, brain contusion and skull fracture, and the patient received emergency surgery.	14 days	Tetraplegia, dysarthria, dysphagia, and respiratory failure.	Increased protein levels.	The latency of F wave was prolonged, and motor nerve block in the upper and lower extremities.

TBI: traumatic brain injury; CSF: cerebrospinal fluid; CMAP: compound muscle action potential.

## Data Availability

The data presented in this study are available upon request from the corresponding author.

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
