# Peer review of "Guillain-Barré Syndrome with Incomplete Oculomotor Nerve Palsy after Traumatic Brain Injury: Case Report and Literature Review"

_brainsci, 2023, doi:10.3390/brainsci13040527_

Round 1

Reviewer 1 Report

this is an interesting study by Liu et al titled Guillain-Barré syndrome with incomplete oculomotor nerve palsy after traumatic brain injury: case report and literature review. The authors discussed a case study of GBS after traumatic brain injury with a 66-year-old male patient who presented during conservative treatment for TBI.

the case of interest however the work fails a lot in the discussion.

the way this case study is written is descriptive and there is a lot of literature where the concept of autoantibody and autoimmunity are discussed.

a good source article for example Autoantibodies in traumatic brain injury and central nervous system trauma, indicates that TBI induces the generation of autoantigens that can lead to GBS and these autoimmunity processes mediated by the B cells can be highly pathological in nature.

I advise the authors to elaborate on the discussion section. they need to discuss that the GBS occurrence can be due to the secondary phase of TBI rather than being sporadically occurring. This is of high importance to clinical treatment as well.

Minor correction:

please use the TBI acronym: instead of traumatic brain injury

Reviewer 2 Report

This manuscript describes a case study of a 66-year-old male patient with Guillian-Barre syndrome (GBS) after presenting with traumatic brain injury (TBI). The authors suggest that GBS should be considered in patients with GBS after TBI.

Line 22: Use the full name of GBS to start the Introduction even though it is mentioned in the abstract.

Line 26-27: PLease add a few sentences with references describing previous research that shows the connection between TBI and GBS. The studies below may be useful.

Carr, K. R., Shah, M., Garvin, R., Shakir, A., & Jackson, C. (2015). Post-Traumatic brain injury (TBI) presenting with Guillain-Barré syndrome and elevated anti-ganglioside antibodies: a case report and review of the literature. The International journal of neuroscience125(7), 486–492. https://doi.org/10.3109/00207454.2014.957760

Zhang, Y., Huang, C., Lu, W., & Hu, Q. (2022). Case Report: Delayed Guillain-Barré syndrome following trauma: A case series and manage considerations. Frontiers in surgery9, 903334. https://doi.org/10.3389/fsurg.2022.903334

Huang, C., Zhang, Y., Deng, S., Ren, Y., & Lu, W. (2020). Trauma-Related Guillain-Barré Syndrome: Systematic Review of an Emerging Concept. Frontiers in neurology11, 588290. https://doi.org/10.3389/fneur.2020.588290

Line 32-33: Change to "His Glasgow Coma Scale score was 12 (eye opening...) and he had no history..."

Line 44: Change to "The patient was transferred..."

Line 62: Change to "Ten days following admission, however, bilateral..."

Line 102: Change to "...on the tenth day..." and change to "..clinicians lack awareness of GBS after TBI."

Line 124: Change to "...extremities and respiratory failure..."

Line 149: This sentence  from "...it is hard to think..." is poorly written and confusing. Please rewrite it. Do you mean it is difficult to establish a time line of causation?

Line 150: Delete "in"

Line 155: Delete "was"

Lines 159-160: Delete "in the future"

Round 2

Reviewer 2 Report

The manuscript is improved and I think it is ready for publication. I just have a few minor suggestions before finalising it.

Line 28: Change "firstly" to "first" and "Subsequent study has reported..." to A subsequent study reported..."

Line 29-30: "...traumatic patients..." to "...trauma patients..." and "...higher risk to develop..." to "...higher risk of developing..." and delete "that"